# Outcomes and factors associated with medical treatment failure in patients with spinal epidural abscess: A 14-year experience

**María García de Santos**[1], **Jorge Calderón-Parra**[1,2]*, **Andrea Gutiérrez-Villanueva**[1,2], **Itziar Diego-Yagüe**[1,2], **Noemí Lomillos Prieto**[3], **Oscar Gil de Sagredo del Corral**[3], **Ana Fernández-Cruz**[1,2,4], **Antonio Ramos-Martínez**[1,2,4], **Elena Muñez-Rubio**[1,2]

**1** Infectious Diseases Unit, Internal Medicine Department, University Hospital Puerta de Hierro, Madrid, Spain, **2** Research Institute Puerta de Hierro-Segovia de Aranda (IDIPHSA), Madrid, Spain, **3** Neurosurgery Department, University Hospital Puerta de Hierro, Madrid, Spain, **4** Medicine Department, Autonomous University of Madrid, Madrid, Spain

* Jorge050390@gmail.com

## Abstract

### Background

The optimal therapeutic approach for treating spinal epidural abscesses (SEAs) is not well defined. This study aimed to describe the failure rate of medical management and identify factors associated with failure.

### Methods

We conducted a single-centre retrospective cohort including all adult patients diagnosed with SEA between 2009 and 2022. The primary endpoint was a composite of in-hospital mortality and motor neurological sequelae at discharge.

### Results

Among 76 patients, 22.4% (n = 17) received initial intervention, while 77.6% (n = 59) received medical management. Among medically managed patients, 42.5% (n = 25) experienced treatment failure, and 27.1% (n = 16) required salvage surgery. Factors associated with treatment failure included diabetes mellitus (32.0% vs 8.8%, p = 0.040), an erythrocyte sedimentation rate (ESR) greater than 75 mm/h (66.7% vs 31.0%, p = 0.021), methicillin-resistant *Staphylococcus aureus* (MRSA) (28.0% vs 0%, p < 0.001) and anterior epidural involvement (91.3% vs 60.6%, p = 0.014). Although patients initially treated surgically had significant worse neurological motor situation at presentation than those managed medically (ASIA A or B 20.0% vs 3.4%), the primary endpoint occurred more frequently in patients with failure of initial medical treatment than in those initially operated (65.0% vs 35.3%, p = 0.035). SEA-related mortality was also higher among those with medical treatment failure (16.0% vs 0%, p = 0.038).

**Data availability statement:** The anonymized dataset underlying the findings described in this paper is fully available in a public repository at URL: https://zenodo.org/records/18496127.

**Funding:** The author(s) received no specific funding for this work.

**Competing interests:** The authors have declared that no competing interests exist.

## Conclusions

Failure of medical management of SEA was common and could lead to worse outcomes. Diabetes mellitus, ESR greater than 75 mm/h, MRSA, and ventral epidural involvement were associated with failure. Initial surgery might be considered for low operative risk patients in the presence of these factors. Prospective trials are needed to better guide initial management strategies.

## Introduction

Spinal epidural abscesses (SEAs) represent rare but serious infections with significant repercussions. Reported mortality rates reach up to 20%, and many survivors experience severe neurological sequelae that impair their quality of life [1,2].The neurological deficits resulting from SEAs can become irreversible within 48 hours of onset [3,4]. Therefore, timely diagnosis and effective treatment are essential to prevent possible complications and achieve adequate resolution, minimizing possible sequelae.

Nonetheless, the optimal therapeutic strategy (only antibiotics vs a combination of antibiotics plus drainage intervention) for these patients is not well defined. There is still controversy about the need for surgery in all patients with SEA [5–7].Historically, SEA was regarded as an absolute indication for drainage intervention [8,9]. However, over the last two decades, small case series and cohorts have reported favourable outcomes in certain patients treated medically (antibiotics only) [10–13]. Consequently, many experts now recommend an initial medical approach in patients without neurological deficits [14,15].

Nevertheless, the current evidence supporting the efficacy and safety of an antibiotic-based treatment for SEA is controversial. Other cohorts have reported a high rate of therapeutic failure with impact in both quality of life and neurological motor sequelae [5,16–21]. Reflecting this uncertainty, a recent Delphi consensus recommendations failed to reach agreement on whether SEA's represent a surgical indication regardless the presence of neurological deficits [22].

In this context, we aimed to describe the rate of clinical failure after initial medical management of patients with SEA and to compare the outcomes of these patients to those who underwent early drainage intervention. We also aimed to identify factors associated with failure after medical treatment.

## Materials and methods

We conducted a single-center retrospective cohort at a tertiary hospital. Our center is a 700-bed university hospital serving a reference population of over 500,000 inhabitants. All types of complex spinal surgery are performed.

We included all adult patients (over 18 years) diagnosed with SEA between January 2009 and December 2022. Patients with isolated SEA and those with associated spondylodiscitis were eligible. There were no exclusion criteria.

Epidemiological, clinical, microbiological, radiological and prognostic variables were obtained from the electronic medical records using an anonymized form. Data management was performed using REDCap electronic capture tools, with licenses provided to the Research Institute Puerta de Hierro-Segovia de Arana [23,24]. The study was approved by the Hospital Puerta de Hierro Ethics Committee (code PI 240/25). Owing to its retrospective design, the requirement for informed consent was waived. The study was conducted in accordance with the recommendations of the Declaration of Helsinki.

## Definitions

Patients included in the study were classified into two mutually exclusive groups according to the initial management intention: medical management (antibiotic treatment only) or interventional management (antibiotic therapy plus drainage procedures, either by open surgery or by interventional radiology). Bone or paravertebral biopsy without drainage of the SEA was not considered an interventional management.

Surgical indications and decisions (including initial surgery and rescue surgeries) were made by the attending neuro-surgery team, in a case-by-case basis, taking into account the clinical and radiological findings.

Medical treatment failure was defined as the appearance of any of the following events in a patient initially treated with antibiotics only: SEA-related mortality, worsening neurological symptoms, radiological progression of the SEA, persistence of fever and/or positive blood cultures after 72 hours of appropriate antibiotic therapy, or significant worsening pain. A single patient could fulfill more than one criterion for treatment failure. The antibiotic regimen used was considered appropriate if it provides *in-vitro* coverage against the microbiological isolate. In the absence of microbiological confirmation, the antibiotic was considered appropriate if the prescription followed current guidelines-recommended regimens [14,15].

For the assessment of neurological involvement, we quantified motor involvement via the American Spinal Injury Association (ASIA) scale, in which involvement is graded from A (complete spinal cord injury) to E (no neurological involvement). Quantification was also carried out via the ASIA motor score (ASIAms), in which a quantitative assessment is made of each major muscle group from 0–5, producing a score from 0 to100.

## Endpoints

The primary endpoint was a composite of in-hospital mortality and the presence of neurological motor sequelae at discharge. The secondary endpoints included in-hospital mortality, SEA-related mortality, neurological motor sequela at discharge and at follow-up, worst ASIA and ASIAms during admission, ASIA and ASIAms at discharge, and variation in ASIAms from admission to discharge and to follow-up.

## Statistical analysis

Qualitative variables were expressed as percentages and absolute frequencies. Quantitative variables were expressed as medians with interquartile ranges (IQRs) or as means with standard deviations (SDs).

For inferential analyses, qualitative variables were compared using the chi2 test (or Fisher's exact test when necessary). Quantitative variables were compared using Student's t test or the Mann–Whitney U test. Owing to the low number of events (deaths), no multivariate analysis was performed.

Bilateral p values lower than 0.05 were considered significant. All analyses were performed using SPSS software, version 25 (IBM, Chicago, USA) was used.

## Results

A total of 76 patients with SEA were included. Of these, 17 (22.4%) received initial intervention, whereas 59 (77.6%) received initial medical management. Among the initial interventions, 15 (88.2%) involved open surgery, and two (11.8%) involved drainage through interventional radiology.

Among medically managed patients, 25 (42.4%) met the criteria for failure. In comparison, treatment failure occurred in 3 patients (17.6%) within initial intervention group (p = 0.053); 1 due to SEA-related death and 2 due to neurological deterioration.

## Spinal epidural abscess description

Table 1 summarizes the baseline demographic, clinical, microbiological and radiological characteristics of the cohort. The median age was 67 years (RIQ 57–76), 35.5% were women (27/76), and the median Charlson comorbidity index was 1 (RIQ 0–3). Twelve patients (15.8%) had prior spinal surgery, and seven (9.2%) had spinal instrumentation.

The median duration of symptoms until diagnosis was 2 weeks (IQR 1–4). Only 52.6% (40) had fever, and 17.1% (13) presented motor neurological impairment. Bacteremia was documented in 52.0% (39/75) of the patients. The most frequent etiologies were *Staphylococcus aureus* (46.1%) and coagulase-negative staphylococci (13.2%), followed by gram-negative bacteria (9.2%) and streptococci (6.6%). A polymicrobial etiology was found in 9.9% of the patients, whereas 21.1% (16) had no microbiological diagnosis.

Regarding radiological findings, SEAs were more common at the lumbar (47.4%) or thoracic levels (44.7%). Only 7.9% of patients had cervical involvement. In 53.4% of the patients, the lesions extended over more than one intervertebral space, contacted the spinal cord in 32% and produced radiological myelitis in 6.7%. A total of 13.2% of the patients had non-contiguous SEAs. In 69.4% of the patients, the abscess involved the anterior epidural area. In 90.2% (69/76) there was concomitant spondylodiscitis

The initial empirical antibiotic was appropriate in 93.2% of the patients. The median duration of antibiotic treatment was 57 days (IQR 45–99). In-hospital mortality was 10.5% (8), with SEA-related mortality accounting for 6 deaths (8.1%).

## Factors associated with initial surgical management

Table 1 compares the characteristics of patients who underwent initial intervention versus those who receiving medical management. Patients with initial intervention had more frequent ASIA B or A at presentation (20% vs 3.4%, p = 0.038), lower ASIAms (89 (SD 21) vs 97 (SD 7), p = 0.021) and higher rates of contact of the abscess with the spinal cord (64.3% vs. 23.7%, p = 0.006). Ventral involvement was more common in the medical group, although the difference did not reach statistical significance (73.2% vs 50.0%, p = 0.094). The identification of methicillin-susceptible *Staphylococcus aureus* and coagulase-negative staphylococci was more common in patients who initially underwent surgery (66.7% vs 30.5%, p = 0.012; and 33.3% vs 8.5%, p = 0.037, respectively). Differences between groups were also found in the prevalence of diabetes mellitus, previous stroke and obesity, presence of bacteremia and the level of spine involvement, as shown in table 1.

## Factors associated with medical treatment failure

Among the patients initially managed medically (n = 59), 25 (42.4%) met criteria for failure. Sixteen patients (27.1% of the total, 64.0% of failures) underwent salvage surgery. The median time to treatment failure was 21 days (RIQ 14–40). The most frequent reasons for failure included radiological progression (n = 14), pain worsening (n = 13) and neurological deterioration (n = 7). The reasons for not performing an initial surgery in those with medical treatment failure were a surgical team's decision according to neurological status and MRI findings in the vast majority (n = 24). Only 1 patient was considered not-suitable for surgery due to prohibitive comorbidity and poor baseline functional and cognitive status.

Of note, 8 patients presented with neurological motor impairment and were not initially operated. Of these, 2 patients had pre-existing motor deficit (1 of them failed medical treatment and were later operated), 2 had a deficit lasting more than 48 hours (1 of them also failed and were later operated), 1 had poor previous functional and cognitive status (died due to SEA) and 3 were classified as "mild" neurological deficit. These 3 patients failed medical treatment.

**Table 1. Univariate comparison of patients with initial intervention versus initial medical treatment.**

| Variable | | Total (n = 76) | Initial intervention (n = 17) | Initial medical management (n = 59) | p |
|---|---|---|---|---|---|
| ***COMORBIDITIES*** | | | | | |
| Sex (female) | | 35.5% (27/76) | 23.5% (4/17) | 39.0% (23/59) | 0.389 |
| Age (years) | | 67.5 (IQR 57–76) | 63 (IQR 52–73) | 69 (IQR 57–76) | 0.133 |
| Arterial hypertension | | 52.6% (40/76) | 70.6% (12/17) | 47.5% (28/59) | 0.079 |
| Diabetes mellitus | | 23.7% (18/76) | 41.2% (7/17) | 18.6% (11/59) | 0.058 |
| Chronic lung disease | | 22.4% (17/76) | 11.8% (2/17) | 25.4% (15/59) | 0.491 |
| Chronic cardiac failure | | 14.5% (11/76) | 5.9% (1/17) | 16.9% (10/59) | 0.235 |
| Chronic renal failure | | 11.8% (9/76) | 17.6% (3/17) | 10.2% (6/59) | 0.322 |
| Chronic liver disease | | 11.8% (9/76) | 5.9% (1/17) | 13.6% (8/59) | 0.353 |
| Active cancer | | 9.2% (7/76) | 11.8% (2/17) | 8.5% (5/59) | 0.494 |
| Previous stroke | | 13.2% (10/76) | 35.3% (6/17) | 6.8% (4/59) | 0.007 |
| Immunosuppression | | 5.3% (4/76) | 0% (0/17) | 6.8% (4/59) | 0.355 |
| Cognitive decline/dementia | | 5.3% (4/76) | 5.9% (1/17) | 5.1% (3/59) | 0.227 |
| Obesity | | 31.6% (24/76) | 52.9% (9/17) | 25.4% (15/59) | 0.034 |
| Age-adjusted Charlson Index | | 3(IQR 2–5) | 2 (IQR 1–4) | 4 (IQR 2–5) | 0.147 |
| Previous spinal surgery | | 15.8% (12/76) | 23.5% (4/17) | 13.6% (8/59) | 0.260 |
| ***CLINICAL MANIFESTATIONS*** | | | | | |
| Symptoms duration (weeks) | | 2 (IQR 1–4) | 2 (IQR 1–4) | 3 (IQR 1–4) | 0.112 |
| Back pain | | 94.7% (72/76) | 94.1% (16/17) | 94.9% (56/59) | 0.645 |
| Fever | | 52.6% (40/76) | 52.9% (9/17) | 52.4% (31/59) | 0.423 |
| Motor function impairment | | 17.1% (13/76) | 29.4% (5/17) | 13.6% (8/59) | 0.126 |
| ASIA score at admission | E | 82.9% (63/76) | 66.7% (10/15) | 86.4% (51/59) | 0.072 |
| | D or C | 10.5% (8/76) | 13.3% (2/15) | 10.2% (6/59) | 0.660 |
| | B or A | 6.6% (5/76) | 20% (3/15) | 3.4% (2/59) | 0.038 |
| C-reactive protein > 100 mg/dL | | 45.9% (34/74) | 50% (7/14) | 43.1% (25/58) | 0.306 |
| ESR greater than 75 mm/h | | 49.2% (31/63) | 63.6% (7/11) | 46% (23/50) | 0.662 |
| Acute renal failure | | 19.7% (15/76) | 23.5% (4/17) | 18.6% (11/59) | 0.445 |
| Septic shock | | 1.3% (1/75) | 0% (0/16) | 1.7% (1/59) | 0.787 |
| **OTHER SEPTIC INVOLVEMENT** | | | | | |
| Infective endocarditis | | 6.6% (5/76) | 6.7% (1/15) | 6.8% (4/59) | 1.000 |
| Pneumonia/empyema | | 11.8% (9/76) | 13.3% (2/15) | 11.9% (7/59) | 1.000 |
| Meningitis | | 5.3% (4/76) | 13.3% (2/15) | 3.9% (2/59) | 0.265 |
| Non-vertebral arthritis | | 13.2% (10/76) | 0% (0/15) | 16.9% (10/59) | 0.187 |
| ***MICROBIOLOGY*** | | | | | |
| Bacteremia | | 52% (39/75) | 75% (12/16) | 45.8% (27/59) | 0.035 |
| *Staphylococcus aureus* | | 46.1% (35/76) | 58.8% (10/17) | 42.4% (25/59) | 0.230 |
| Methicillin-susceptible | | 36.8% (28/76) | 58.8% (10/17) | 30.5% (18/59) | 0.012 |
| Methicillin-resistant | | 9.2% (7/76) | 0% (0/17) | 11.9% (7/59) | 0.450 |
| Coagulase negative staphylococci | | 13.2% (10/76) | 29.4% (5/17) | 8.5% (5/59) | 0.037 |
| *Streptococcus spp.* | | 6.6% (5/76) | 5.9% (1/17) | 6.8% (4/59) | 1.000 |
| *Mycobacterium tuberculosis* | | 5.3% (4/76) | 6.7% (1/15) | 5.1% (3/59) | 1.000 |
| Other | | 19.7% (11/76) | 6.6% (1/15) | 16.9% (10/59) | 0.443 |
| No known etiology | | 21.1% (16/76) | 11.8% (2/17) | 23.7% (14/59) | 0.286 |
| Polymicrobial etiology | | 9.9% (7/71) | 17.6% (3/17) | 7.1% (4/56) | 0.158 |

*(Continued)*

**Table 1.** (Continued)

| Variable | | Total (n = 76) | Initial intervention (n = 17) | Initial medical management (n = 59) | p |
|---|---|---|---|---|---|
| *RADIOLOGY* | | | | | |
| Higher radiological involvement | Cervical | 7.9% (6/76) | 26.7% (4/15) | 3.4% (2/59) | 0.048 |
| | Dorsal | 44.7% (34/76) | 33.3% (5/15) | 47.5% (28/59) | |
| | Lumbar | 47.4% (36/76) | 40% (6/15) | 49.2% (29/59) | |
| More than 1 space involvement | | 53.4% (39/73) | 57.1% (8/14) | 50.9% (29/57) | 0.547 |
| Ventral epidural involvement | | 69.4% (50/72) | 50% (7/14) | 73.2% (41/56) | 0.094 |
| Non-contiguous abscesses | | 13.2% (10/76) | 26.7% (4/15) | 10.2% (6/59) | 0.573 |
| Medullary contact | | 32% (24/75) | 64.3% (9/14) | 23.7% (14/59) | 0.006 |
| Radiological myelitis | | 6.7% (5/75) | 21.4% (3/14) | 3.4% (2/59) | 0.169 |
| *MANAGEMENT* | | | | | |
| Adequate empirical ATB | | 93,2% (69/74) | 87,5% (14/16) | 94,8% (55/58) | 0.579 |
| Intravenous ATB (days) | | 21 (14-42) | 20 (14-28) | 21 (14-37) | 0.630 |
| Oral sequential ATB | | 84,0% (63/75) | 81,3% (13/16) | 84,7% (50/59) | 1.000 |
| Total ATB duration | | 57 (45-99) | 59 (43-117) | 56 (45-98) | 0.547 |
| Admission duration (days) | | 36 (IQR 22–57) | 35 (14-52) | 36 (22-62) | 0.379 |

Table 2 compares the characteristics of patients who were successfully managed medically and those who failed medical treatment. Patients with failure more frequently presented diabetes mellitus (32.0% vs 8.8%, p = 0.040), a erythrocyte sedimentation rate (ESR) greater than 75 mm/h (66.7% vs 31.0%, p = 0.021), methicillin-resistant *Staphylococcus aureus* (MRSA) isolation (28.0% vs 0%, p = 0.001) and ventral epidural involvement (91.3% vs 60.6%, p = 0.014). There were no differences in other baseline characteristics, clinical presentation, microbiology or radiology between groups.

Although rates of adequacy of empirical antibiotics were similar between groups, patients with medical treatment failure had longer intravenous antibiotic durations (26 days vs. 15, p = 0.033) and longer hospital admissions (46 days vs. 29, p = 0.012).

## Primary and secondary endpoints

Table 3 shows the outcomes according to treatment group. The composite primary endpoint occurred significantly more frequently in patients who failed medical treatment (65.0%, 13/20) than in those without medical treatment failure (12.5%, 4/32, p < 0.001) or those treated initially with surgery (35.3%, 6/17, p = 0.035).

Patients with medical treatment failure, compared with those without failure and those who initially underwent surgery, had higher SEA-related mortality (16.0% vs 0% vs 5.9%, p = 0.038). Neurological sequelae at discharge were also more common (31.3% vs. 6.3% vs. 25.0%, p = 0.013). Additionally, patients with medical treatment failure had worse ASIA scores at discharge (87 (SD 30) vs 99 (SD 2) vs 93 (SD 27), p = 0.035). They also showed greater deterioration in ASIAms from admission to follow-up (−3.8 (SD 6.5) vs + 0.6 (SD 2.1) vs + 4.8 (SD 8.5), p = 0.051).

## Discussion

In our work, we aimed to describe the failure rate of medical management for SEAs and to evaluate the impact of such failure on clinical outcomes. Our main findings indicate that the failure rate of medical treatment is high (up to 40%) and could be associated with worse outcomes. In our cohort, factors associated with treatment failure included diabetes mellitus, an elevated ESR, MRSA isolation and ventral involvement of the abscess. Accordingly, initial surgery may be considered in those patients with these risk factors. In cases in whom the decision was made to treat only with antibiotic management, close monitoring during the first weeks is highly recommended.

In our series, three-quarters of the patients were initially managed without drainage intervention. This proportion is higher than that reported in both historical and recent series [25–27]. Nevertheless, this trend is consistent with the most

**Table 2.** Univariate comparison of patients with initial medical management with and without treatment failure.

| Variable | | Medical treatment (n = 59) | Successful medical treatment (n = 34) | Failure medical management (n = 25) | p |
|---|---|---|---|---|---|
| ***COMORBIDITIES*** | | | | | |
| Sex (female) | | 38.9% (23/59) | 38.2% (13/34) | 40.0% (10/25) | 1.000 |
| Age (years) | | 67.5 (IQR 57–76) | 73 (IQR 57–79) | 66 (IQR 59–75) | 0.472 |
| Arterial hypertension | | 47.5% (28/59) | 55.9% (19/34) | 36.0% (9/25) | 0.188 |
| Diabetes mellitus | | 18.6% (11/59) | 8.8% (3/34) | 32.0% (8/25) | 0.040 |
| Chronic lung disease | | 25.4% (15/59) | 14.7% (5/34) | 20.0% (5/25) | 0.087 |
| Chronic cardiac failure | | 16.9% (10/59) | 14.7% (5/34) | 20.0% (5/25) | 0.729 |
| Chronic renal failure | | 10.2% (6/59) | 5.9% (2/34) | 16.0% (4/25) | 0.386 |
| Chronic liver disease | | 13.6% (8/59) | 14.7% (5/34) | 12.0% (3/25) | 1.000 |
| Active cancer | | 8.5% (5/59) | 8.8% (3/34) | 8.0% (2/25) | 1.000 |
| Previous stroke | | 6.8% (4/59) | 8.8% (3/34) | 4.0% (1/25) | 0.630 |
| Immunosuppression | | 6.8% (4/59) | 8.8% (3/34) | 4.0% (1/25) | 0.630 |
| Cognitive decline/dementia | | 1.7% (1/59) | 2.9% (1/34) | 0% (0/25) | 1.000 |
| Obesity | | 25.4% (15/59) | 23.5% (8/34) | 28.0% (7/25) | 0.767 |
| Age-adjusted Charlson Index | | 3 (2-5) | 4 (2-5) | 3 (2-6) | 0.827 |
| Previous spinal surgery | | 13.6% (8/59) | 11.8% (4/34) | 16.0% (4/25) | 0.711 |
| ***CLINICAL MANIFESTATIONS*** | | | | | |
| Symptoms duration (weeks) | | 2 (IQR 1–4) | 2.5 (IQR 1–4) | 3 (IQR 1–4) | 0.863 |
| Back pain | | 94.9% (56/59) | 91.2% (31/34) | 100% (25/25) | 0.255 |
| Fever | | 52.5% (31/59) | 52.9% (18/34) | 52.0% (13/25) | 1.000 |
| Motor function impairment | | 13.6% (8/59) | 8.8% (3/34) | 20.0% (5/25) | 0.347 |
| Sensitive function impairment | | 10.2% (6/59) | 5.9% (2/34) | 16.0% (4/25) | 0.424 |
| ASIA score at admission | A | 86.4% (51/59) | 91.2% (31/34) | 80.0% (20/25) | 0.681 |
| | D or C | 10.2% (6/59) | 5.9% (2/34) | 16.0% (4/25) | |
| | B or A | 3.4% (2/59) | 2.9% (1/34) | 4.0% (1/25) | |
| C-reactive protein > 100 mg/dL | | 43.1% (25/58) | 33.3% (11/33) | 56.0% (14/25) | 0.111 |
| ESR (mm/h) | | 72 (IQR 42–106) | 62 (IQR 35–96) | 91 (IQR 64–107) | 0.022 |
| ESR greater than 75 mm/h | | 46% (23/50) | 31.03% (9/29) | 66.7% (14/21) | 0.021 |
| Acute renal failure | | 18.6% (11/59) | 11.8% (4/34) | 28.0% (7/25) | 0.176 |
| Septic shock | | 1.7% (1/59) | 0% (0/34) | 4.0% (1/25) | 0.424 |
| **OTHER SEPTIC INVOLVEMENT** | | | | | |
| Infective endocarditis | | 6.8% (4/59) | 8.8% (3/34) | 4.0% (1/25) | 0.630 |
| Pneumonia/empyema | | 11.9% (7/59) | 11.8% (4/34) | 12.0% (3/25) | 1.000 |
| Meningitis | | 3.4% (2/59) | 0% (0/34) | 8.0% (2/25) | 0.175 |
| Non-vertebral arthritis | | 16.9% (10/59) | 11.8% (4/34) | 24.0% (6/25) | 0.297 |
| ***MICROBIOLOGY*** | | | | | |
| Bacteremia | | 45.8% (27/59) | 41.2% (14/34) | 52.0% (13/25) | 0.440 |
| *Staphylococcus aureus* | | 42.4% (25/59) | 32.4% (11/34) | 56.0% (14/25) | 0.069 |
| Methicillin-susceptible | | 30.5% (18/59) | 32.4% (11/34) | 28.0% (7/25) | 0.781 |
| Methicillin-resistant | | 11.9% (7/59) | 0% (0/34) | 28.0% (7/25) | 0.001 |
| Coagulase negative staphylococci | | 8.5% (5/59) | 5.9% (2/34) | 12.0% (3/25) | 0.641 |
| *Streptococcus spp.* | | 6.8% (4/59) | 5.9% (2/34) | 8.0% (2/25) | 1.000 |
| *Mycobacterium tuberculosis* | | 5.1% (3/59) | 2.9% (1/34) | 8.0% (2/25) | 0.569 |
| Other | | 16.9% (10/59) | 20.6% (7/34) | 12.0% (3/25) | 0.493 |

*(Continued)*

**Table 2.** (Continued)

| Variable | | Medical treatment (n = 59) | Successful medical treatment (n = 34) | Failure medical management (n = 25) | p |
|---|---|---|---|---|---|
| No known etiology | | 23.7% (14/59) | 35.3% (12/34) | 8% (2/25) | 0.028 |
| Polymicrobial etiology | | 7.1% (4/56) | 6.3% (2/32) | 8.3% (2/24) | 1.000 |
| *RADIOLOGY* | | | | | |
| Higher radiological involvement | Cervical | 3.4% (2/59) | 5.9% (2/34) | 0.0% (0/25) | 0.166 |
| | Dorsal | 47.5% (28/59) | 38.2% (13/34) | 60.0% (15/25) | |
| | Lumbar | 49.2% (29/59) | 55.9% (19/34) | 40.0% (10/25) | |
| More than 1 space involvement | | 50.9% (29/57) | 47.1% (16/34) | 56.5% (13/23) | 0.592 |
| Ventral epidural involvement | | 73.2% (41/56) | 60.61% (20/33) | 91.3% (21/23) | 0.014 |
| Non-contiguous abscesses | | 10.2% (6/59) | 11.8% (4/34) | 8.0% (2/25) | 0.921 |
| Medullary contact | | 23.7% (14/59) | 17.6% (6/34) | 32.0% (8/25) | 0.229 |
| Radiological myelitis | | 3.4% (2/59) | 0% (0/34) | 8.0% (2/25) | 0.175 |
| *MANAGEMENT* | | | | | |
| Adequate empirical ATB | | 94,8% (55/58) | 97,0% (32/33) | 92,0% (23/25) | 0,572 |
| Intravenous ATB duration (days) | | 21 (14-37) | 15 (13-31) | 26 (21-46) | 0.033 |
| Oral sequential ATB | | 84,7% (50/59) | 91,2% (31/34) | 76,0% (19/25) | 0.149 |
| Total ATB duration | | 56 (45-98) | 56 (44-82) | 70 (45-147) | 0.279 |
| Admission duration (days) | | 36 (22-62) | 29 (17-50) | 46 (33-74) | 0,012 |

recent recommendations from international societies, that propose a trial of medical management in neurologically intact patients [14,15]. A recent meta-analysis revealed that medical management is becoming increasingly common [6]. However, these recommendations are largely based on small retrospective studies involving selected patients [11–13]. Thus, current evidence on the safety of this approach is limited. Indeed, some experts continue to favour early surgical management for these infections [22,28]. This is reflected in a recent Delphi consensus, in which 63% of experts recommended that "An epidural empyema is almost always an indication for surgery, even in the absence of neurological deficit" [22].

A key insight from our study is the high rate of therapeutic failure among patients initially treated with antibiotics alone. The rate of failure to medical management is similar to was found in a recent systematic review [16]. Additionally, similarly high rates have been described previously in a number of works [5,6,17,29–31]. Although almost all patients in our cohort received appropriate initial antibiotics, 40% experienced therapeutic failure in the first 3 weeks, and a significant proportion required surgical rescue.

Importantly, failure of medical management can have significant consequences for patients [5,21,32]. In our cohort, patients who failed antibiotic treatment had significantly worse outcomes than those who underwent initial surgery did (frequency of primary endpoint 65.0% vs. 35.3%, p = 0.035). This worse prognosis was due mainly to higher SEA-related mortality and a higher frequency of motor sequelae. Hospital admissions were also longer in those with failure. Other recent studies have reached similar conclusions to our study [5,20,21,32]. Notably, a meta-analysis published in 2023 with involving more than 10,000 patients with spondylodiscitis (with and without SEAs) showed reduced mortality associated with early surgery compared with initial medical management [19].

Despite these data, the risks and morbidities associated with surgical intervention should not be forgotten [33–35]. In our series, we observed that the majority of patients (almost 60%) actually experienced a favorable clinical outcome with medical management. Therefore, we believe that it is of vital important to define those factors associated with higher risk of failure. In our study, we found that diabetes mellitus, an ESR greater than 75 mm/h, ventral epidural involvement and

**Table 3. Comparison of prognosis and outcomes among patients with initial intervention, successful medical treatment and failure to medical treatment.**

| Outcome variable | | Initial surgical treatment (n = 17) | Successful medical treatment (n = 34) | Failure medical management (n = 25) | p |
|---|---|---|---|---|---|
| Composite primary endpoint | | 35.3% (6/17) | 12.5% (4/32) | 65.0% (13/20) | 0.001 |
| In-hospital mortality | | 5.9% (1) | 5.9% (2) | 20.0% (5) | 0.122 |
| *Abscess-related mortality* | | 5.9% (1) | 0 | 16.0% (4) | 0.038 |
| *Non-related mortality* | | 0 | 5.8% (2) | 4.0% (1) | 1.000 |
| Worst ASIAms | | 87.8 (SD 19.7) | 97.7 (SD 7.4) | 91.4 (SD 13.7) | 0.020 |
| Worst ASIA | E | 50.0% (8/16) | 84.8% (28/33) | 56.0% (14/25) | 0.019 |
| | D-C | 25.0% (4/16) | 12.2% (4/33) | 32.0% (8/25) | |
| | B-A | 25.0% (4/16) | 3.0% (1/33) | 12.0% (3/25) | |
| Neurological sequalae at discharge | | 31.3% (5/16) | 6.3% (2/32) | 25.0% (8/20) | 0.013 |
| ASIAms at discharge | | 92.6 (SD 26.7) | 99.7 (SD 1.7) | 87.0 (SD 30) | 0.035 |
| ASIAms from admission to discharge | | 2.6 (SD 7.9) | 1.3 (SD 4.8) | −1.8 (SD 13.3) | 0.322 |
| ASIAms alteration at discharge | | 21.4% (3/14) | 3.2% (1/31) | 30% (5/20) | 0.037 |
| ASIA at discharge | E | 78.6% (11/14) | 93.5% (29/31) | 75.0% (15/20) | 0.152 |
| | D-C | 21.4% (3/14) | 6.5% (2/31) | 20.0% (4/20) | |
| | B-A | 0 | 0 | 5.0% (1/20) | |
| Neurological sequala at follow-up | | 21.4% (3/14) | 3.2% (1/31) | 27.8% (5/18) | 0.024 |
| ASIAms at follow-up | | 92.1 (SD 27.7) | 96.8 (SD 17.9) | 89.0 (SD 30.7) | 0.059 |
| ASIAms from admission to follow-up | | 4.8 (SD 8.5) | 0.6 (SD 2.1) | −3.8 (SD 6.5) | 0.051 |
| ASIAms alteration at follow-up | | 23.1 (3/13) | 3.3% (1/30) | 25.0% (5/20) | 0.058 |
| ASIA at follow-up | E | 83.3% (10/12) | 96.7% (29/30) | 88.2% (15/17) | 0.166 |
| | D-C | 16.7% (2/12) | 3.3% (1/30) | 5.9% (1/17) | |
| | B-A | 0 | 0 | 5.9% (1/17) | |

MRSA were associated with higher rates of failure. All these factors have been previously associated with medical treatment failure by other authors [5,6,28,30–32,36]. Patel et al [5] described higher rates of failure in patients with diabetes mellitus and very high acute phase reactants, whereas Spernovasilis et al [36] found that MRSA was associated with medical treatment failure. The aforementioned factors have biological plausibility that explains their association with worse outcomes. Hyperglycemia present in patients with diabetes and immunological impairments reported with this disease could favor abscess growth [37]. Likewise, a greater degree of analytical alteration (whether measured by ESR or C-reactive protein) could be associated with greater infection burden [5]. In ventral abscesses, medical treatment is likely to be offered more frequently than in posterior abscesses because of the technical difficulty to access to this location. Finally, the antibiotics commonly used in MRSA infections (vancomycin or daptomycin) have poor penetration of the blood–brain barrier, [36,38]. Another important factor may be the level of involvement, with worse outcomes reported in patients with cervical abscesses [33,38]. In our cohort, we were unable to demonstrate this association, perhaps due to the small number of patients with cervical involvement.

Taken together, these observations support an individualized patient-centred approach. Early surgery may be considered for patients with low operative risk who present with factors associated with medical treatment failure, even when neurological deficits are absent [3,7,22,27]. On the basis of several (but not all) of the previously mentioned factors, there has been proposed a web-based application that can inform the probability of medical treatment failure using artificial intelligence-based algorithms [31]. Nevertheless, more studies are needed to refine the risk stratification. Specifically, prospective and randomized trials are needed to prove the potential benefit of early surgery in neurologically intact patients with risk factors for failure.

This study provides important insights into SEAs management, although it is not without limitations. This was a single-center retrospective study, with inherent limitations due to its design. First, our relatively small sample size may have limited the analysis of factors associated with failure. Nevertheless, our results are consistent with recent literature. Second, we could not analyze morbidities associated with surgery, especially those that were not clearly measurable (i.e., postoperative pain). However, the lower mortality and motor sequelae observed in this group support the hypothesis of its safety in selected low-risk patients. Third, we could not determine the exact timing of the neurological deficit at presentation, which could have influenced the absence of surgery indications in patients with motor neurological deficits. Yet, recent studies have shown that patients with SEAs could experience neurological improvement after surgery even with established neurological compromise (more than 24–48 hours) [39,40]. Accordingly, a longer duration of neurological deficit onset may not be a categorical contraindication to surgery in SEAs. Finally, the observational nature of our work prevents us from drawing robust conclusions and to recommend specific treatment pathways. Yet, our study supports the need for an early risk stratification and poses the base for possible future research.

In conclusion, the failure rate of initial medical treatment for SEAs was high and could lead to worse outcomes. Factors associated with failure included diabetes mellitus, an ESR greater than 75 mm/h, MRSA isolation, and involvement of the ventral area of the epidural space. Initial surgical management could be considered in patients with these factors and low operative risk. For those medically managed patients, close clinical and radiological monitoring should be advised. Prospective randomized studies are required to better define initial management strategies of patients with SEA.

## Author contributions

**Conceptualization:** Jorge Calderón-Parra, Elena Muñez-Rubio.

**Data curation:** María García de Santos, Jorge Calderón-Parra.

**Formal analysis:** María García de Santos, Jorge Calderón-Parra.

**Investigation:** Jorge Calderón-Parra.

**Methodology:** Jorge Calderón-Parra.

**Project administration:** Jorge Calderón-Parra, Elena Muñez-Rubio.

**Supervision:** Elena Muñez-Rubio.

**Writing – original draft:** María García de Santos, Jorge Calderón-Parra.

**Writing – review & editing:** Jorge Calderón-Parra, Andrea Gutiérrez-Villanueva, Itziar Diego-Yagüe, Noemi Lomillos Prieto, Oscar Gil de Sagredo del Corral, Ana Fernández-Cruz, Antonio Ramos-Martínez, Elena Muñez-Rubio.

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
