## [Decision Letter · Decision Letter 0]

21 Jan 2026

PONE-D-25-58296Outcomes and factors associated with conservative treatment failure in patients with spinal epidural abscess. A 14-year experience.PLOS One

Dear Dr. Calderón-Parra,

Thank you for submitting your manuscript to PLOS ONE. After careful consideration, we feel that it has merit but does not fully meet PLOS ONE’s publication criteria as it currently stands. Therefore, we invite you to submit a revised version of the manuscript that addresses the points raised during the review process.

We look forward to seeing your revision, addressing the issues raised by the reviewers.

We look forward to receiving your revised manuscript.

Kind regards,

Rodney Adam

Academic Editor

PLOS One

2. In the online submission form, you indicated that [All relevant data are within the manuscript and its supporting information files. Additional data, including patient's database, is available upon reasonable request].

Reviewers' comments:

Reviewer's Responses to Questions

**Comments to the Author**

1. Is the manuscript technically sound, and do the data support the conclusions?

Reviewer #1: Yes

Reviewer #2: Yes

2. Has the statistical analysis been performed appropriately and rigorously? 

Reviewer #1: No

Reviewer #2: Yes

3. Have the authors made all data underlying the findings in their manuscript fully available?

Reviewer #1: No

Reviewer #2: No

4. Is the manuscript presented in an intelligible fashion and written in standard English?

Reviewer #1: No

Reviewer #2: Yes

5. Review Comments to the Author

Reviewer #1: 1. Use of terms: There is inconsistency in use of terms in the manuscript, particularly conservative and medical management. Medical management is a more appropriate and informative term and should be used throughout.

2. Grammar: There is frequent use of incorrect words - like unicentre (instead of single center), use of worse instead of worst in table 3 and spelling errors (e.g. bacteriemia, sequalea). Syntax is frequently inappropriate. The manuscript needs a thorough review and English language editing.

3. Microbiology: 59 patients underwent medical management, of whom only 27 had bacteremia. Yet a microbiologic diagnosis was found in all but 14 patients (45 patients). I note that 16 subsequently underwent salvage surgery- assuming all had positive cultures from the procedure (which is unlikely), that adds up to 43. How was a microbiologic diagnosis arrived in those who had negative blood cultures and did not get any drainage done?

4. Adequacy of antibiotic therapy: 55/58 medically managed patients were deemed to have adequate empiric antibiotics yet only 45 had a known microbiologic diagnosis. How is this possible?

5. Statistical analysis and tables: The denominators in table 2 remain 34 and 25 throughout. However, the eligible patients for that parameter are fewer e.g. total 11 patients with diabetes. Is it more appropriate to use the number of eligible patients with that characteristic to be the denominator rather than all patients in that group? E.g. for patients with diabetes - 3/11 had successful medical management and 8/11 failed medical management.

There are some places where the numbers do not add up. An example is table 3 where 2 patients who had successful medical management had in hospital mortality. However in the breakdown only 1 patient (with non-abscess related mortality) appears. What happened to the other patient?

The tables need thorough review for consistency

6. Comparison of treatment failure: It is not mentioned how many patients who had an initial intervention had treatment failure. This would be useful information for comparison

7. Success of medical management with no known microbiologic etiology: How do you explain that majority (12/14) patients with no known microbiologic etiology (hence treated empirically) had successful medical management?

Reviewer #2: Comments to the Authors

This is a well-structured retrospective single-center study analyzing risk factors for failure of conservative treatment in patients with spinal epidural abscesses (SEA). The authors provide a dataset from a 14-year period, including clinical parameters, treatment pathways, and outcomes. Overall, the study adds valuable information to the ongoing debate on optimal management of SEAs.

However, please elaborate on the following comments for improvement:

On prior literature: Please discuss the findings of Stratton et al. (JNS Spine, 2016), who analyzed similar predictors of failure in conservative management of SEA. Their work might serve as a benchmark for comparison with your data.

On Patel et al. (Spine J, 2014): This paper was included and cited, but a more direct comparison of your main results with theirs (especially regarding risk factors for surgical conversion) would strengthen the discussion.

On terminology and differentiation: Please address whether patients with isolated epidural abscesses without spondylodiscitis were included and, if so, how you ensured diagnostic accuracy. This distinction is important for the generalizability of your conclusions.

On recent consensus: The 2025 EANS Consensus Statement on de novo spinal infections (Brain and Spine) should be cited. In particular:

Statement #15: “An epidural empyema is almost always an indication for surgery, even in the absence of neurological deficit” received no consensus.

Statement #16: “An epidural empyema causing a neurological deficit is a clear indication for decompressive surgery” reached 100% agreement.

According to your data, some patients with neurological deficits were still treated conservatively. Please clarify how this aligns with or challenges the consensus recommendation.

On surgical indication thresholds: The discussion would benefit from a clearer outline of your institutional criteria for switching to surgery. Was the decision based solely on clinical deterioration, or were imaging findings also decisive?

On implications for practice: While the retrospective nature limits the ability to recommend specific treatment pathways, your data may support refinement of early risk stratification models. This point could be made more explicit.

Minor editing: Please ensure consistency in terminology (e.g., SEA vs. spinal empyema), and double-check grammar in the abstract and discussion.

6. PLOS authors have the option to publish the peer review history of their article (what does this mean?). If published, this will include your full peer review and any attached files.

Reviewer #1: No

Reviewer #2: **Yes:** Andreas Kramer

---

## [Author Response · Author response to Decision Letter 1]

17 Feb 2026

Estimated reviewers,

We tank you for your suggestions which surely have improved the overall quality of the manuscript. Please find attached the point-by-point respone:

Reviewer #1:

1. Use of terms: There is inconsistency in use of terms in the manuscript, particularly conservative and medical management. Medical management is a more appropriate and informative term and should be used throughout.

Response: Thank you for the suggestion. We have changed the term.

2. Grammar: There is frequent use of incorrect words - like unicentre (instead of single center), use of worse instead of worst in table 3 and spelling errors (e.g. bacteriemia, sequalea). Syntax is frequently inappropriate. The manuscript needs a thorough review and English language editing.

Response: We have made the suggested modification and the manuscript has been revised by an native English speakers to improve the syntaxis and language.

3. Microbiology: 59 patients underwent medical management, of whom only 27 had bacteremia. Yet a microbiologic diagnosis was found in all but 14 patients (45 patients). I note that 16 subsequently underwent salvage surgery- assuming all had positive cultures from the procedure (which is unlikely), that adds up to 43. How was a microbiologic diagnosis arrived in those who had negative blood cultures and did not get any drainage done?

Response: Out of the 59 patients with initial medical management, 27 were diagnosed by blood cultures. Of the 32 patients remaining, in 18 cases diagnosis was made by culture of local sample, including 7 in the rescue surgery (out of 16) and 11 by bone or paravertebral biopsy (not abscess punction and drainage, which would have been considered a intervention). Of note, among the patients without diagnosis (n=14), in 9 cases a bone biopsy was also obtained, but with negative cultures. We have now clarified this point in the text.

4. Adequacy of antibiotic therapy: 55/58 medically managed patients were deemed to have adequate empiric antibiotics yet only 45 had a known microbiologic diagnosis. How is this possible?

Response: Thank you for raising this point. Indeed, we acknowledge that we did not specify the definition of adequate antibiotic in the text. We have now added this definition. To clarify your question, in the absence of microbiology diagnosis, antibiotic was considered adequate if the scheme was in line with current guideline recommendations (added as reference as well)

5. Statistical analysis and tables: The denominators in table 2 remain 34 and 25 throughout. However, the eligible patients for that parameter are fewer e.g. total 11 patients with diabetes. Is it more appropriate to use the number of eligible patients with that characteristic to be the denominator rather than all patients in that group? E.g. for patients with diabetes - 3/11 had successful medical management and 8/11 failed medical management.

Response: Thank you for the suggestion. I acknowledge that there is an advantage in the way you represent the numerator/denominator: 72.3% (8/11) of patients with diabetes failed to medical treatment and only 27.7% of patients with diabetes succeeded it. However, this number does not include the percentage of patients without diabetes who failed, which is necessary to state that a certain characteristic is a factor associated with failure. In this example, of the patients without diabetes (n=48), 17 presented failure (35.4%), the p-value is still the same (p=0.040) because the contingency table for the Fisher test is the same.

However, although we can write or represent this information in graphics, we would have to due a separate graphic for each variable, which is not feasible.

For this reason, we represented in the table the percentage of patients with and without medical failure with a certain characteristic: diabetes, hypertension, immunosuppression, etc. This allowed to asses if that characteristic is a factor associated with failure by comparing the percentage of patients with that characteristic in both groups. For instance, in your example, diabetes mellitus was present in 32.0% of patients with failure (9/25) versus 8.8% of patients without failure (3/34), p=0.040. It is the same comparison as previously, with the same information, but it allows us to summarize all the information in one table.

There are some places where the numbers do not add up. An example is table 3 where 2 patients who had successful medical management had in hospital mortality. However in the breakdown only 1 patient (with non-abscess related mortality) appears. What happened to the other patient?

The tables need thorough review for consistency

Thank you for noticing the mistake. We have checked that both patients died due to unrelated reasons (pulmonary embolism and cardiac failure). Actually, the percentage (5.9%) was correct, which correspond to 2 patients, no 1. We have revised the rest of the tables for consistency

6. Comparison of treatment failure: It is not mentioned how many patients who had an initial intervention had treatment failure. This would be useful information for comparison

Response: We have added this information to the text. However, treatment failure criteria were selected with medical treatment in mind, and were not developed for surgically treated patients. Of the criteria used to define failure, in those patients with initial surgery, 1 presented SEA-related death, 2 presented worsening of neurologic symptoms. NO patient presented persistent fever/bacteremia, radiological worsening or significant worsening of pain. This represent 3 patients 17.6% (3/17), which was fewer patients that in initially medical management, but without statistically significance (p=0.053)

7. Success of medical management with no known microbiologic etiology: How do you explain that majority (12/14) patients with no known microbiologic etiology (hence treated empirically) had successful medical management?

Response: The probable explanation for this is related to your third question (microbiology). The variable of etiology was not determined initially but at the end of the process. Many patients with treatment failure underwent salvage surgery, with corresponding cultures, which could explain that the majority of patients with unknown etiology had successful medical treatment. Additionally, in patients with good clinical evolution it is less probable that treatment physician pursues a biopsy for culture.

It would have been interesting to address if knowing the etiology at the beginning (by means of blood culture or early bone culture) is a factor associated with success of medical treatment. Unfortunately, we did not specify this variable in the original dataset, so we could not analyse this factor. We can say that positive blood culture was not associated with medical treatment outcomes.

Reviewer #2: Comments to the Authors

This is a well-structured retrospective single-center study analyzing risk factors for failure of conservative treatment in patients with spinal epidural abscesses (SEA). The authors provide a dataset from a 14-year period, including clinical parameters, treatment pathways, and outcomes. Overall, the study adds valuable information to the ongoing debate on optimal management of SEAs.

However, please elaborate on the following comments for improvement:

On prior literature: Please discuss the findings of Stratton et al. (JNS Spine, 2016), who analyzed similar predictors of failure in conservative management of SEA. Their work might serve as a benchmark for comparison with your data.

Response: We thank the reviewer for the interesting reference. Indeed, this systematic review yielded a similarly high rate of medical treatment failure. We have added the reference and discussed it in the text. However, they did not analyse predictors of failure, neither if that failure led to worse clinical outcome.

On Patel et al. (Spine J, 2014): This paper was included and cited, but a more direct comparison of your main results with theirs (especially regarding risk factors for surgical conversion) would strengthen the discussion.

Response: We thank the reviewer for the suggestion. We have added a direct comparison of risk factors.

On terminology and differentiation: Please address whether patients with isolated epidural abscesses without spondylodiscitis were included and, if so, how you ensured diagnostic accuracy. This distinction is important for the generalizability of your conclusions.

Response: We did included patients with spinal epidural abscess, whether or not they had concomitant spondylodiscitis. We have added this information to the text. Of note, out of the 76 included patients, 69 (90.8%) had spondylodiscitis and only 9.2% did not, as expected. The diagnosis of the spondylodiscitis was made in accordance to radiological findings. We admit that it can be argue that a bone or disc involvement would be present in virtually all patients with SEA, whether or not seen in radiological test.

On recent consensus: The 2025 EANS Consensus Statement on de novo spinal infections (Brain and Spine) should be cited. In particular:

Statement #15: “An epidural empyema is almost always an indication for surgery, even in the absence of neurological deficit” received no consensus.

Statement #16: “An epidural empyema causing a neurological deficit is a clear indication for decompressive surgery” reached 100% agreement.

Response: We thank the reviewer for the reference. We reviewed the literature previously to this reference was published, and, later, we did not see it. Yet, we acknowledge that this is a very important reference for the text. We have added the reference and discussed it in the text.

According to your data, some patients with neurological deficits were still treated conservatively. Please clarify how this aligns with or challenges the consensus recommendation.

Response: Indeed, in our series there were 8 patients with neurological deficits that did not undergo initial surgery. Of them, in 2 the deficit was previous to the infection, other 2 were diagnosed > 48 hours from clinical presentation, and in 1 had severe cognitive impairment (the patient who did not undergo surgery due to poor previous functional status). In the rest (3) the decision was not to operate. In retrospective review of the chart, the reason not to operate was that the deficit was “mild” (the described muscular balance was 4/5).

Of these patients: the 3 patients with “mild” impairment were later operate as were considered to have failure. The 1 patient with severe cognitive impairment died. Of the 2 patients with neurological deficit >48h, 1 had failure and operation and 1 had treatment success. Finally, of the 2 patients with previous neurological deficit, 1 succeeded medical treatment and the other did not.

To clarify, we do not defend that a “mild” neurological deficit is not a surgical indication. This was only a real-world practice. We have added this information to the text.

On surgical indication thresholds: The discussion would benefit from a clearer outline of your institutional criteria for switching to surgery. Was the decision based solely on clinical deterioration, or were imaging findings also decisive?

Response: In our institution, there is not a specific protocol on when to operate a SEA. The decision is made in a case-by-case basis. This includes the rescue surgery in case of treatment failure. As per many surgeries and institution, in also depends on the specific surgeon responsible for the patient. We have clarified this in the text.

On implications for practice: While the retrospective nature limits the ability to recommend specific treatment pathways, your data may support refinement of early risk stratification models. This point could be made more explicit.

Response: Thank you for the suggestion. We have strengthened the importance of risk stratification.

Minor editing: Please ensure consistency in terminology (e.g., SEA vs. spinal empyema), and double-check grammar in the abstract and discussion.

Response: We have checked the consistency of different terminology, as also suggested by other reviewer. The manuscript has been revised by an native English speaker to improve the gramar and language

---

## [Decision Letter · Decision Letter 1]

30 Mar 2026

PONE-D-25-58296R1Outcomes and factors associated with conservative treatment failure in patients with spinal epidural abscess. A 14-year experience.PLOS One

Dear Dr. Calderón-Parra,

Thank you for submitting your manuscript to PLOS ONE. After careful consideration, we feel that it has merit but does not fully meet PLOS ONE’s publication criteria as it currently stands. Therefore, we invite you to submit a revised version of the manuscript that addresses the points raised during the review process.

**Please note the comments from reviewer 1 and respond to those comments.**

We look forward to receiving your revised manuscript.

Kind regards,

Rodney Adam

Academic Editor

PLOS One

Journal Requirements:

Reviewers' comments:

Reviewer's Responses to Questions

**Comments to the Author**

1. If the authors have adequately addressed your comments raised in a previous round of review and you feel that this manuscript is now acceptable for publication, you may indicate that here to bypass the “Comments to the Author” section, enter your conflict of interest statement in the “Confidential to Editor” section, and submit your "Accept" recommendation.

Reviewer #1: (No Response)

Reviewer #2: All comments have been addressed

2. Is the manuscript technically sound, and do the data support the conclusions?

Reviewer #1: Yes

Reviewer #2: Yes

3. Has the statistical analysis been performed appropriately and rigorously? 

Reviewer #1: Yes

Reviewer #2: Yes

4. Have the authors made all data underlying the findings in their manuscript fully available?

Reviewer #1: Yes

Reviewer #2: Yes

5. Is the manuscript presented in an intelligible fashion and written in standard English?

Reviewer #1: No

Reviewer #2: Yes

6. Review Comments to the Author

Reviewer #1: 1. Conservative management still appears throughout the manuscript instead of medical management. As noted previously medical management is a better term.

2. "Patients with medical treatment failure, compared with those without failure and those

who initially underwent surgery, had higher SEA-related mortality (16.0% vs 5.9% vs

0%, p=0.038)." This statement contradicts with "In comparison, 3 patients in the initial intervention group

presented criteria for failure (17.6%, p=0.053); 1 due to SEA-related death and 2 due

to neurological worsening" when it comes to SEA related mortality.

3. "There were no other differences in clinical presentation, radiology, microbiology or antibiotic management between the groups." As per the table, other significant differences between the not mentioned in the text include previous stroke, obesity, level of spine involvement and presence of bacteremia. Kindly review this.

4. There are a lot of spelling and grammatical errors littered through the document. The English is not standard. This needs to be addressed.

Reviewer #2: The authors have adequately addressed all comments raised in the previous review round. In particular, the clarification regarding the management of patients with neurological deficits significantly improves the transparency and clinical interpretation of the study. The inclusion of relevant literature and current consensus recommendations has strengthened the discussion. The manuscript is technically sound, the data are clearly presented, and the conclusions are supported by the results. I have no further comments and support publication.

7. PLOS authors have the option to publish the peer review history of their article (what does this mean?). If published, this will include your full peer review and any attached files.

Reviewer #1: No

Reviewer #2: **Yes:** Andreas Kramer

---

## [Author Response · Author response to Decision Letter 2]

31 Mar 2026

Response to reviewers:

Reviewer #1:

1. Conservative management still appears throughout the manuscript instead of medical management. As noted previously medical management is a better term.

Response: We have revised the manuscript and changed the term conservative to medical as suggested.

2. "Patients with medical treatment failure, compared with those without failure and those

who initially underwent surgery, had higher SEA-related mortality (16.0% vs 5.9% vs

0%, p=0.038)." This statement contradicts with "In comparison, 3 patients in the initial intervention group presented criteria for failure (17.6%, p=0.053); 1 due to SEA-related death and 2 due to neurological worsening" when it comes to SEA related mortality.

Response: Thank you for noticing the error. It was a mistake in writing the sentence. The correct percentages of SEA-related mortality were: 16.0% (4/25) of those with failure to medical treatment, vs 0% (0/34) of those without failure and 5.9% (1/17) of those with initial surgery, as also shown in table 3. The way the sentence was written, it should state: “"Patients with medical treatment failure, compared with those without failure and those who initially underwent surgery, had higher SEA-related mortality (16.0% vs 0% vs 5.9%, p=0.038).". We have changed the sentence.

3. "There were no other differences in clinical presentation, radiology, microbiology or antibiotic management between the groups." As per the table, other significant differences between the not mentioned in the text include previous stroke, obesity, level of spine involvement and presence of bacteremia. Kindly review this.

Response: You are correct. We have modified the sentence.

4. There are a lot of spelling and grammatical errors littered through the document. The English is not standard. This needs to be addressed.

Response. Thank you for the suggestion. We have now corrected the spelling and grammar of the manuscript according to recommendations of a native English speaker

Reviewer #2:

The authors have adequately addressed all comments raised in the previous review round. In particular, the clarification regarding the management of patients with neurological deficits significantly improves the transparency and clinical interpretation of the study. The inclusion of relevant literature and current consensus recommendations has strengthened the discussion. The manuscript is technically sound, the data are clearly presented, and the conclusions are supported by the results. I have no further comments and support publication.

Response: Thank you for your revision and the previous comments to the paper. We agree that the response to the comments have improved the quality of the manuscript.

---

## [Editor Report · Decision Letter 2]

15 Apr 2026

Outcomes and factors associated with medical treatment failure in patients with spinal epidural abscess. A 14-year experience

PONE-D-25-58296R2

Dear Dr. Calderón-Parra,

We’re pleased to inform you that your manuscript has been judged scientifically suitable for publication and will be formally accepted for publication once it meets all outstanding technical requirements.

Kind regards,

Rodney Adam

Academic Editor

PLOS One
---

## [Editor Report · Acceptance letter]

PONE-D-25-58296R2

PLOS One

Dear Dr. Calderón-Parra,

I'm pleased to inform you that your manuscript has been deemed suitable for publication in PLOS One. Congratulations! Your manuscript is now being handed over to our production team.

Kind regards,

on behalf of

Dr. Rodney Adam

Academic Editor

PLOS One